# Debt Collection Model for Mass Receivables Based on Decision Rules—A Path to Efficiency and Sustainability

**Rafał Jankowski \*** and **Andrzej Paliński \***

Faculty of Management, AGH University, 30-059 Kraków, Poland
* Correspondence: rjankowski@agh.edu.pl (R.J.); palinski@agh.edu.pl (A.P.)

**Abstract:** Debt collection companies buy overdue debts on the market in order to collect them and recover the highest possible amount of a debt. The pursuit of debt recovery by employees of collection agencies is a very demanding task. The aim of the article is to propose a rule-based model for managing the process of mass debt collection in a debt collection company, which will make the debt collection process more efficient. To achieve this, we have chosen a decision tree as a machine learning technique best suited for creating rules based on extensive data from the debt collection company. The classification accuracy of the decision tree, regardless of the possibility of acquiring rule-based knowledge, proved to be the highest among the tested machine learning methods, with an accuracy rate of 85.5%. Through experiments, we generated 16 stable rules to assist in the debt collection process. The proposed approach allows for the elimination of debts that are difficult to recover at the initial stage of the recovery process and to decide whether to pursue amicable debt collection or to escalate the debt recovery process to legal action. Our approach also enables the determination of specific actions during each stage of the proceedings. Abandoning certain actions or reducing their frequency will alleviate the burden on collection agency employees and help to avoid the typical burnout associated with this line of work. This is the path to making the organizational culture of a collection agency more sustainable. Our model also confirms the possibility of using data from debt collection companies to automatically generate procedural rules and automate the process of purchasing and collecting debts. However, this would require a larger set of attributes than what we currently possess.

**Keywords:** debt collection company; debt trading; mass debt; employee sustainability; job burnout; sustainable organization; decision tree; machine learning; business rules

## 1. Introduction

During economic turnover, obligations arise for sold goods or services as well as public law fees. These obligations are not always repaid on time by the debtor, or not repaid at all. In such a situation, the creditor may attempt to recover the debt themselves or sell it at a discount to a specialized debt recovery firm. Debt collection companies purchase overdue debts in order to recover as much as possible from them. The process of collecting overdue debts is usually lengthy, involves many stages, and does not always result in repayment of a debt at a satisfactory level.

The debt collection process requires collection agency employees to have multiple contacts with debtors to enforce debt repayment. These contacts are often very psychologically demanding and quickly lead to burnout, resulting in short employment durations. Avoiding unnecessary contact for debts with low recovery chances and reducing the frequency of contact when the nature of the debt allows it are methods to make work in a collection agency more sustainable. Awareness of the debt collection company's responsibility for employees can lead to their greater job engagement and reduce the risk of burnout [1].

An additional motivation for undertaking this research is the need to evaluate the usefulness of machine learning methods in generating decision rules from debt data and to explore the possibilities of automating the debt recovery process.

One type of activity conducted by debt collection companies is the purchase of mass debts of small amounts. A mass debt portfolio usually comprises telephone subscriptions, energy charges, purchase of access to streaming services, etc. Managing the portfolio of purchased debt packages is one of the key management areas in a debt collection company. The portfolio of debts is built on the basis of the selection of debt packages available on the market. The proper debt recovery process influences the financial result of a debt collection company.

The models of the debt collection process used in practice do not always lead to proper recovery of a receivables portfolio. This state of affairs is due to not fully utilizing the knowledge contained in the historical data on the debt collection process by a given collection company. In our opinion, the solution to the problem is the use of machine learning methods to discover dependencies in historical datasets. The use of knowledge discovery methods, followed by automation of the debt purchasing and collection process, is a current necessity, as the European debt purchase market approached 25 billion EUR in 2020 and has been growing at double-digit rates [2].

The main goal of our research was to build a model for the collection process of mass receivables using machine learning techniques based on the characteristics of receivables. To achieve our main goal, we formulated the following research questions.

Q1. Is it possible to construct a set of decision rules for the debt collection process based on the data held by a debt collection company?

Q2. Will machine learning methods allow for the creation of non-trivial rules of conduct consistent with expert knowledge?

Q3. Will machine learning methods provide new insights into the debt collection process, leading to the elimination of unnecessary actions?

Q4. Will it be possible to automate the debt collection process based on discovered decision rules and an inference engine?

As part of the research, a number of machine learning methods were compared and, consequently, a decision tree was selected as a tool to discover the relationships between the characteristics of debts and the level of recovery obtained through various procedural methods in the historical data. These dependencies took into account both the characteristics of debts and the features of debtors themselves. The use of a decision tree also made it possible to generate decision rules understandable to experts conducting the collection process of receivables.

The further part of the article is organized as follows. Section 2 presents the procedures and legal foundations of the debt collection process using the example of Poland, as well as a review of global research in the field of debt collection. Section 3 provides a brief formal introduction to the methodology of acquiring decision rules in the machine learning process. Section 4 outlines the characteristics of the data used in the study and the methodology for building a machine learning model for generating rules. Section 5 presents the research results, including a list of generated debt collection rules and a proposed new model for managing the process of mass debt collection. The article concludes with a discussion of the results and a summary of the research, along with suggestions for further research directions.

## 2. Debt Collecting Methodology and Related Works

### 2.1. Mass Debt Collection—The Case of Poland

The purchase of mass debt portfolios is carried out by specialized debt collection entities. Such purchases may involve debts at various stages of delinquency and originating from different creditors. The traditional model of debt portfolio acquisition by a debt collection entity includes [3]:

- Selection of a debt portfolio from those available on the market;
- Analysis of the debt portfolio to estimate the associated risk of purchasing the portfolio, usually based on a random sample drawn from the entire portfolio;
- Valuation of the debt portfolio based on factors such as the risk level of debt non-payment, the number and distribution of nominal values of the debts being sold, the debtor's organizational structure, the completeness of source documentation, and the statute of limitations;
- Negotiations regarding the acquisition terms, including the price of the debt portfolio, payment forms and schedule, and acceptable debt collection techniques (if the seller imposes such requirements during the bidding process).

Subsequent stages include: signing a debt assignment agreement, transferring data, payment for the debts according to the agreed schedule, initiating the preparation process, and then debt collection according to the procedures adopted by the debt collection entity.

The turnover of mass debts is not limited solely to the purchase of debt portfolios. Existing forms of turnover evolve with the development of the market toward, for example, factoring or debt exchange, constituting a secondary market for the turnover of bulk debts.

Acquiring a mass debt portfolio initiates the debt collection process in the debt collection entity, the optimization of which is the aim of this research. Regardless of the adopted collection model, the process of debt recovery can be divided into stages. The basic division includes two stages of debt collection [4]:

- Amicable debt collection;
- Compulsory (enforcement, court) debt collection.

In the case of amicable debt collection, the activities undertaken do not involve the use of legal coercion, unlike court proceedings and enforcement actions, which constitute the basis of the compulsory stage of debt collection. This division is one of the basic ones, but a more comprehensive one distinguishes 3–4 stages of the debt collection process, namely, the amicable, judicial, enforcement, and post-enforcement stages [5]. The debt collection process in banks involves: early monitoring, late monitoring, pre-litigation debt collection, litigation debt collection, enforcement proceedings, and debt sale [6]. In our research, we have adopted a two-stage division, treating the legal debt collection stage and enforcement (execution) as one stage—enforcement debt collection.

Below is an attempt to briefly characterize the stages of debt collection, along with indicating possible actions and collection techniques.

Amicable debt collection is a process that involves negotiations with the debtor aimed at voluntary settlement of the debt. Voluntary settlement by the debtor does not involve as high costs as the judicial route and is less time-consuming [7].

The aim of the amicable debt collection process is both to recover the debt and to prepare for any subsequent actions or stages of debt collection. At this stage, missing documents are supplemented, and information about the debtor is obtained and updated. Such actions are also taken in an automated manner, creating connections based on data such as tax identification number, insurance number, address, contact phone number, and linking cases and debts under one debtor.

Actions taken in the amicable debt collection stage require the use of tools aimed at persuading the debtor to settle the arrears. These tools include: telephone debt collection, reminder letters, formal demand letters, pre-litigation demand letters, field collector visits, reminder notices (SMS, email), public disclosure of debt information, debt sale, and interest calculation and collection. They are usually applied in various combinations. The scope of their application and the order will depend on the adopted debt collection path, debt collection procedures, or the classification of debts serviced in the mass model [8].

Signing agreements with the debtor at the amicable debt collection stage aims not only to establish a repayment schedule but also to confirm the acknowledgment of the obligation, which is significant in the event of the debtor breaching the agreement and consequently initiating legal proceedings. Taking punitive actions at the amicable debt collection stage involves charging and collecting penalty interest on overdue obligations or

public disclosure of debt. The most commonly used form of debt disclosure is registration in existing debtor registers in a given country.

In the case of mass debts, tools supporting the debt collection process include [7]: electronic payment identification, automatic generation and sending of correspondence. The importance of automating the generation and sending of correspondence increases with the number of cases (debts) handled, resulting in cost reduction, shorter preparation time for correspondence, and easier verification of its correctness. Conducting debt collection at the amicable stage, in accordance with social norms and legal regulations, is the primary way of recovering overdue debts. It is important to note the increasing significance of cultural differences, which must be taken into account by debt collection entities. The importance of this aspect of debt collection grows when conducting debt collection concerning different cultural groups, for example, as a result of conducting debt collection in international markets. Also crucial is the market value of assets owned by the debtor, as it determines the debtor's willingness to negotiate and repay [9].

The compulsory debt collection stage includes successive actions related to court debt collection and enforcement (enforcement). In compulsory debt collection, the creditor resorts to legal instruments to secure and enforce the amounts owed, which may be a continuation of the unsuccessful amicable debt collection stage or a deliberate choice of the creditor. At the judicial debt collection stage, the creditor may pursue claims according to the legal regulations of a given country [10].

The general model of the judicial debt collection process follows the following scheme [11]:

- Initiation of proceedings;
- Obtaining an enforcement title;
- Obtaining an enforcement order.

Obtaining an enforcement order opens the possibility of proceeding to the enforcement (execution) stage. The execution of actions by the court bailiff is carried out only at the request of the creditor and is not automatically initiated upon the issuance of the enforcement order. The creditor can only influence the actions of the bailiff and exercise control over the actions taken to a limited extent. In the case of mass debt collection, it is common practice to build an indicator based on which the effectiveness of bailiffs is evaluated in order to direct debts to bailiffs with the highest effectiveness indicator.

The proposed sequence of the debt collection process is presented in Figure 1. The theoretical model illustrates the entire debt collection process, including stages often overlooked by other researchers, such as the purchase of debts included in the debt portfolio (stage 1), data enrichment in the SKIP tracing process (stage 2), and the selection of collection strategies (stage 3). Each of these stages, despite not being actual debt collection, impacts its effectiveness and the level of recovery generated. Additionally, our theoretical model has been supplemented with a closing stage of the process (stage 6), which introduces actions on the remaining debts, such as their preparation for sale.

The key task served by our research is to replace the third stage currently conducted based on the internal procedures of the debt collection company with an automatic decision-making process based on a decision engine and a rule base. The rule database can be created and updated based on data collected by the debt collection company during the debt collection activities conducted on previously purchased debts.

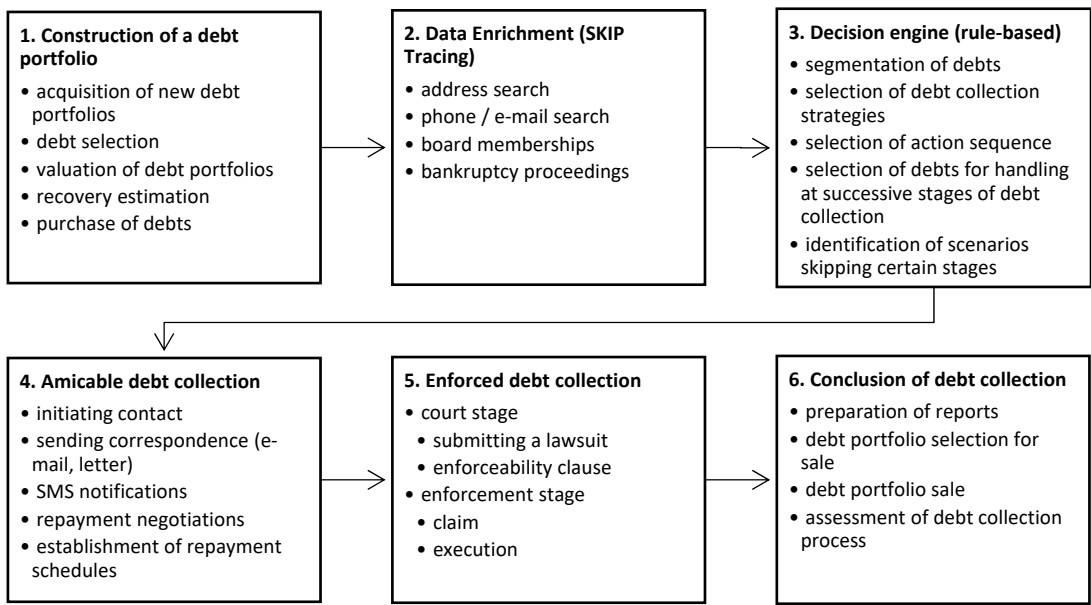

**Figure 1.** Theoretical model of the debt collection process.

*2.2. Related Works*

Research results regarding debt collection by debt collection companies are not very common in the literature. These studies can be classified into two main areas. The first one is the classification of debts into those that can be recovered and those that are unlikely to be repaid. The second area of research concerns the valuation of debts and the schedule of their future repayment. However, there are no studies that aim to determine a detailed model of debt handling by a debt collection company. For example, Santos et al. [12] attempted to find the best machine learning algorithm to predict the success rate in the amicable debt collection process of debts originating from private schools in Brazil. Pinheiro et al. [13] used machine learning methods to recommend (classification problem) amicable proceedings instead of resorting to court proceedings.

A very recent study on the use of machine learning and artificial intelligence methods in human resource management was conducted by the international team of Xiang et al. [14]. Based on extensive data from multiple countries and numerous enterprises, they attempted to determine the impact of artificial intelligence and digital transformation on the sustainable development of employee lifecycle management.

Sanches et al. [15], based on data from Chilean financial institutions, attempted to determine the likelihood of success in three tasks of debt collection process: establishing contact with the debtor, obtaining a promise of repayment, and actual repayment of overdue debts. They used several machine learning methods for this purpose. Using the three-SHAP method, they determined the impact of explanatory variables on the probabilities of success for the three debt collection tasks they analyzed.

On the other hand, Kribel and Yam [16] demonstrated in their research that debt collection companies play a very important role in obtaining information about debtors. Recovery rates increase when additional debtor information is gathered by debt collection companies compared to recovery rates without additional data collected by debt collection companies. The additional information included: spatial information, external credit assessments, customer relationship information, and information on financial and nonfinancial assets.

Furthermore, Geer et al. [17] focused on the time-consuming component of the amicable debt collection process, which is making phone calls to debtors. They optimized the procedure for making phone calls by adjusting the frequency or discontinuing contacts depending on the characteristics of the debt. They significantly improved the debt collection

procedure compared to the traditional method using a uniform scheme for making phone calls to debtors.

Sancarlos et al. [2] used machine learning techniques to calculate the propensity to pay (PtP). The calculated probability of debt repayment allowed for a decision on further debt collection actions in the amicable debt collection process or for referring the debt to court. These studies only determined the probability of repayment but did not specify detailed debt collection procedures. These studies offer the possibility of application in the debt portfolio valuation process.

The selection of explanatory variables in the process of building a debt collection model may include a more or less extensive set of attributes. For example, Pinheiro et al. [13] used the type of debt occurrence, debt situation, stage of debt process, debt balance, protest office, date of occurrence, date of registration, and irregularity (missing/wrong data). Kribel and Yam [16] applied in their study exposure (amount in euros), age of the debtor, a dummy variable for insolvent accounts, a dummy variable for a corporation, a dummy variable for availability of telephone contact, and age of the account.

Features describing the receivables usually include [18]: the class of receivables (in terms of value), the reason for the arrears, the overdue period, the type of document confirming the receivables, and the scope of debt collection activities. In the case of debtor characteristics, a further division is made—another set of characteristics applies to natural persons, and a different set applies to business entities. Examples of personal debtor characteristics for natural persons are: gender, age, place of residence, and education. In the group of debtor businesses, an exemplary set of their characteristics may include: legal form, type of business, place of business, and other payment arrears. Taking into account the characteristics of both debts and debtors in the valuation process is based on the assumption that specific features translate into the level of debt repayment. The proposed debtor features are illustrated in Figure 2.

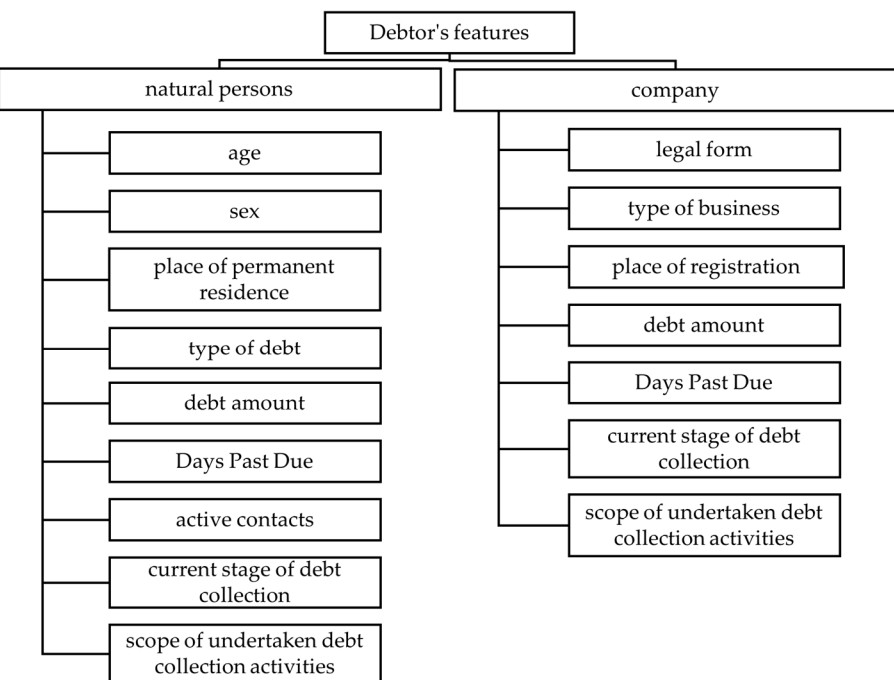

**Figure 2.** Debtor's and debt characteristics.

## 3. Acquisition of Decision Rules in the Process of Machine Learning

### 3.1. Induction of Decision Rules

The knowledge contained in the available datasets, utilized in the management process of a debt collection entity, can be represented in the form of decision rules. Each rule is described in the form of an implication consisting of attributes and one of the possible

decision alternatives. The attributes form the left-hand side of the implication, while the decision alternative forms the right-hand side. Inducing decision rules requires determining the representation method of the data, which constitutes a set of training examples. Such a set is often presented as an information table (*IT*) or a decision table [19]. From a formal perspective, an information table represents a set of pairs, given as follows:

$$IT = (U, A) \tag{1}$$

where:

$U$—a non-empty and finite set of objects;
$A$—a non-empty and finite set of attributes.

To describe the objects, attributes of nominal, ordinal, and numerical types can be used. Introducing a set of decision classes $K$ such that $K = \{K_j: j = 1, \ldots, r\}$, considered in a supervised learning process as an additional attribute, leads to the creation of a decision table (*DT*), as follows:

$$DT = (U, A \cup \{d\}) \tag{2}$$

where:

$d$—a decision attribute not belonging to $A$.

The elements of set $A$ are called conditional attributes. In the case where the decision table *DT* contains examples of concepts, then for each $x \in U$, the value of the information function will be given by $f(x, d) \in K$.

In such a case, a decision rule $r$ describing $K_j$ is defined by the expression:

$$\text{if } P \text{ then } Q \tag{3}$$

where:

$P$—the premise of the rule;
$Q$—the conclusion.

The premise $P$ is also called the antecedent, while the conclusion $Q$ is the decision part of the rule. In some works, decision rules are written in the form $P \rightarrow Q$ or $r$: (Condition)→(Conclusion), where the condition consists of a conjunction of tests on attributes, and the conclusion represents the decision class. The conditional part $P$ consists of composing elementary condition $w_i$ in the following form [20]:

$$P = w_1 \wedge w_2 \wedge w_3 \wedge \ldots \wedge w_m \tag{4}$$

where:

$w_i$—elementary condition;
$m$—the length of the rule (number of conditions).

By elementary condition $w_i$ of rule $r$, we mean the following dependency:

$$(f(\alpha_i, x)) \propto term(\alpha_i)) \tag{5}$$

where:

$\alpha_i$—attribute;
$x$—object;
$f(\alpha_i, x)$—the value of attribute $\alpha_i$ assigned to object $x$;
$term(\alpha_i)$—a constant representing the value from the domain of $\alpha_i$ (elementary term);
$\propto$—relation operator from the set $\{<, \leq, >, \geq, =, \neq, \in\}$.

For the conjunction of elementary conditions, coverage is also determined in a specific decision table as a set of objects from the *DT* (decision table) that satisfy the elementary conditions expressed by $P$. This coverage can be divided into two parts: positive and negative.

The problem of inducing decision rules can be classified as NP-complete [21], and the process itself boils down to finding the minimum set of rules that cover the set of examples [22]. Additionally, the obtained set of rules should enable correct classification of new examples. Generating a set of rules is performed according to an adopted algorithm, based on generating the subsequent coverage of the training set (set of examples). The general procedure leading to the generation of a rule set can be presented as recurring actions: learning a single rule, removing cases covered by the rule, and restarting learning from scratch on the remaining training set (examples). The generation of a single rule should be guided by the principle that the rule should cover as many positive examples as possible while covering the fewest negative examples. Each of the obtained rules is subject to further refinement by adding elementary condition $w_i$. This process is repeated until the accepted conditions for accepting the decision rule are met. Subsequent rules are sought as long as there are positive examples in the set of examples that have not been covered by other obtained rules. Some algorithms alternatively allow for the possibility of early termination of the search for a decision rule [23], e.g., when coverage of the set is found, e.g., in 95% of cases.

From a historical perspective, the first algorithm realizing the idea of generating coverages was the AQ algorithm [20]. This algorithm was further developed in the form of algorithms such as AQ11, AQ15, and AQ17. Another class of algorithms used for inducing decision rules consists of algorithms combining the idea of generating coverages with techniques used for inducing decision trees. The combination of these two solutions was necessitated by the need to consider noisy information (examples). The first example of implementing such an approach was the CN2 algorithm [23,24]. The family of algorithms enabling the induction of decision rules can also be expanded to include algorithms based on rough sets.

Due to the small number of literature items related to the application of machine learning in relation to the debt market, a literature review mainly covered applications related to financial markets. As a result, the scope of research was narrowed down to knowledge discovery methods based, among others, on artificial neural networks, associations, clustering, and decision trees. Artificial neural networks are used in predicting debt repayment [25,26], estimating credit risk [27], fraud detection [28], forecasting stock indices [29], and many other tasks. Unfortunately, artificial neural networks operate in a "black box" model, which makes impossible the creation of rules defining the sequence of debt collection actions.

We verified the possibility of using association rule mining methods in the context of mass debt collection based on an experiment involving multiple attempts to generate rules for the existing database. This led to the construction of rules that are obvious, resulting from legal regulations (e.g., indicating the necessity of sending a payment demand before filing a lawsuit) or have no business application.

Another of the considered approaches was a group of methods related to decision trees. A literature review demonstrated the usefulness of decision trees and derivative methods, including customer classification [30], credit risk assessment [31], and determining the frequency of contact between a debt collector and a debtor [32].

As a result, a decision tree was chosen as a tool enabling the construction of procedural rules and the classification of debts into classes indicating the level of debt repayment ability. The use of decision trees (classification trees) in rule construction has the advantage that the rule conditions directly result from successive splits in the decision tree nodes. The use of decision trees, which are a model of the so-called "white box," additionally reveals the structure of splits and their course, creating the possibility of justifying the procedure when recommending the choice of debt classification in the valuation or debt collection process. Our further research confirmed that the decision tree also exhibited the highest level of classification accuracy compared to the results obtained with other selected methods (SVM, k-NN, and ANN). We present the results of comparing the accuracy levels of individual methods in the subsequent part of the article.

### 3.2. Decision Tree Classifier

Decision trees are one of the most popular groups of algorithms used in the classification problem. Despite the creation of many implementations over the last few years, e.g., ID3, C4.5, CART, and CHAID, all of these classifiers have similar structural schemes and general principles of operation [33]. An important element that distinguishes them is the criterion by which splits are made in successive nodes.

The purpose of classification trees is to divide the provided dataset into smaller, more homogeneous groups. Homogeneity in this case means striving to ensure that in each division node is a proportional excess of observations of one of the output classes over the other. The algorithm searches among the set of attributes for the one value for which division according to that given value will bring the most information in the node. The effect of splitting a node is the creation of a new node for which the increment of information or the final leaf determining the membership in a given class will be recalculated. The following rules determine the completion of the process of generating successive nodes and the formation of a leaf:

- All (or almost all) observations in a node belong to one class;
- There are no further attributes on the basis of which a further division of data can be made;
- The tree has reached its predetermined maximum depth.

In the ID3 and C4.5 algorithms, the division is determined by the cleanliness of a node, which is determined using the information gain indicator. This is a function that maximizes the difference between the purity of the node before and after splitting. The most popular indicator used to directly determine the purity of a given node is entropy. Entropy is defined as follows:

$$Entropy(X) = -\sum_i p_i log_2(p_i) \tag{6}$$

where:

$X$—the attribute for which the entropy is calculated;
$p_i$—the proportion of observations belonging to class $i$.

For the assumed division of $S$, which divides the training set $T$ into several smaller subsets, the weighted sum of the entropy of individual subsets is the average demand for information $H$ and is described by the following formula:

$$H_S(T) = -\sum_{i=1}^{k} P_i H_S(T_i) \tag{7}$$

The value of $H$ is used to determine the information gain that can be obtained as a result of the division of the training set $T$, based on the possible division of $S$.

$$gain(S) = H(T) - H_S(T_i) \tag{8}$$

The division in the decision node with the highest information gain is selected by the C4.5 algorithm as optimal [34].

The CART (Classification And Regression Trees) algorithm [35] uses the following formula as a criterion for the optimal division (for a possible division of $s$ in node $t$):

$$\phi(s|t) = 2P_L P_P Q(s|t) \tag{9}$$

where $P_L$ and $P_P$ describe the ratio of the number of descendants (left and right branches) to the number of the entire training set, and $Q(s|t)$ shows the quantitative difference in the sub-treetops for each value of the target variable. The indicator $Q$ can take the theoretical maximum value of $k$, which is equal to the number of classes of the target variable. The optimal division is the one that achieves the highest value of the $\phi$ index.

For this purpose, the CART algorithm uses the Gini index, which can be interpreted as a criterion for minimizing the probability of incorrect classification [36]. It reaches the minimum (0) when all the observations in the node belong to one category. The Gini index is expressed by the following formula:

$$\text{GiniIndex}(X) = 1 - \sum_{i=1}^{c} p_i(i|t)^2 \tag{10}$$

where:

$p_i(i|t)$—the ratio of class $t$ instances among the training instances in the $i$-th node;
$c$—number of classes.

Learning decision trees create decision rules that are used to infer about future, unknown observations. One rule means any path from root to leaf. The advantage of this solution over other algorithms is ease interpretation. The rules are built in the form of if predecessor then successor, "if—then" [37]. The weakness of decision trees is their susceptibility to abrupt changes in tree structure as a result of changes in input data [38,39].

Our goal was not only to find the most effective machine learning algorithm in the debt classification process, but above all to find rules for the classification of receivables based on the results of the recovery process. For this reason, a decision tree generating rules was recognized as the basic tool for assessing the results of the debt recovery process. These rules can be conveyed directly to the employees of the debt collection company or implemented in an automated decision-making system.

## 4. Materials and Methods

### 4.1. Data

The aim of the research was to build a model for handling mass debts in the form of a rule base. The study was conducted in three stages. In the first stage, interviews were conducted with experts responsible for the debt collection process in debt collection entities. This group comprised six individuals representing six distinct economic entities, including: two owners of debt collection entities, two individuals directly responsible for the debt collection process, one operational director, and one board member.

The characteristics of receivables identified during the interviews were compared with the characteristic features of debts contained in the available data. As a result, the data were modified and prepared for use by data mining methods applied in the second, main stage of the research. The identification of characteristic features of debts, the results of in-depth interviews, and the knowledge discovered in the available dataset formed the basis for building the model of mass debt collection process. Subsequently, in the third stage, the in-depth interview method was again used to validate the constructed rule-based model and to make corrections to it based on expert feedback.

The data used in the study came from four debt collection companies belonging to a capital group operating continuously in the Polish market since 2001. The original dataset contained information on 1,054,237 debts with a total nominal value exceeding PLN 2.7 billion (current approximate exchange rate: 4 PLN = 1 USD) from the period between 2006 and 2022. For the purposes of the study, this set was reduced to 879,007 observations (debts). The reduction of the original dataset resulted from a series of actions aimed at both improving data quality and preparing it for use with selected machine learning methods. As part of pre-processing, incomplete data and outliers were removed. When analyzing the original data, we also examined whether there was an imbalance in the number of observations in each category. In selected cases, debt payment dates were also adjusted to ensure that the first payment date was not earlier than the date of debt package purchase (this phenomenon sometimes occurs when the seller sets a "cut-off date" against which all debtor payments are credited towards the price paid by the debt package buyer). The data provided by debt collection entities for research purposes were transformed into the required format for the applied algorithms, largely undergoing discretization, e.g., debt

amounts were discretized so that they are represented in the model as classes covering debts in specified intervals. Similarly, data on residential areas were categorized based on postal codes. Additionally, we narrowed down the dataset to debts purchased within serviced securitization funds from 2010 to 2021 to standardize them.

Debt amounts were included in nominal values. In our research, we assumed that the debt repayment amount would be compared to the purchase price of the receivable. It turned out that, in 45% of cases, the last installment of the debt was repaid within one year, and in another 22% of cases, it was repaid within the next year after the purchase of the receivable. Thus, the majority of the total repayment amount of purchased receivables occurred in the first year after purchase. The discounting process would therefore not significantly change the proportion of amounts recovered in relation to the purchase prices of receivables.

The accuracy of the data prepared for the mass debt management model was verified by six experts and formed the basis for model corrections. Attributes considered in our research encompassed both debt characteristics and debtor characteristics. During the data preparation process, the original data were transformed or discretized. Based on the dataset available, 10 variables were obtained:

- Account class: Created as a result of discretization of the amount of receivables, respectively in PLN: (0, 500], (500, 1k], (1k, 2k], (2k, 5k], (5k, 10k], (10k, 25k], (25k, 50k], (50k,100k], (100k, 250k], (250k, 500k], and (500k, 1000k].
- Legal form: The attribute assumes one of 10 states—eight storing the legal form under which business activity can be conducted in Poland, and two states of the attribute reserved for indicating natural persons not conducting business activity or in case of missing data.
- Gender: Two states and NA for missing or incorrect data.
- Age: The age of the debtor or their legal representative in the case of a company at the time of the creation of the receivable, rounded to full years, and NA for missing or erroneous data.
- Region: Information about the geographic region of the debtor or the company's headquarters, containing 11 values—10 geographic region codes in Poland and NA for missing or incorrect data.
- Phone call: A binary attribute created as a result of discretization of the number of initiated phone calls by the debt collection entity. In cases where at least one call was made, the value YES (True) was adopted; otherwise, NO (False).
- email: A binary attribute created as a result of discretization of the number of emails sent by the debt collection entity. The attribute took the value YES (True) if an email was sent; otherwise, NO (False).
- Letter: A binary attribute created as a result of discretization of the number of physical letters sent by traditional mail by the debt collection entity. The attribute took the value YES (True) if at least one letter was sent traditionally; otherwise, NO (False).
- SMS: A sent SMS message, the attribute took two states: YES (True) if at least one SMS message was sent; otherwise, NO (False). The class was considered in the previous version of the model but was removed due to minimal information transfer, as in every case (receivable), where a mobile phone number was available, at least one SMS message was sent.
- Collection stage: Information about the stage at which the debt collection was completed. The attribute assumed two states: amicable or enforcement.

A concise description of the variables is presented in Table 1.

In the sample, natural persons accounted for 66% of cases, and enterprises accounted for the remaining 34%. Among natural persons, women accounted for 46% of cases. The amicable recovery stage occurred in 38.5% of cases, and legal enforced recovery occurred in the remaining cases. The vast majority—43.7%—consisted of receivables with small amounts up to PLN 500.

In Tables A1–A4 attached, we present a detailed breakdown of receivables into classes along with selected characteristics of the receivables. Table A5 presents the basic statistics for the selected attributes before discretization.

**Table 1.** Characteristics of variables used in the study.

| Variable | Characteristics of the Variable |
| --- | --- |
| Debt class | Primary debt value class in PLN: (0, 500], (500, 1k], (1k, 2k], (2k, 5k], (5k, 10k], (10k, 25k], (25k, 50k), (50k, 100k], (100k, 250k], (250k, 500k], and (500k, 1000k] * |
| Legal form | Economic Activity, Civil Partnership, General Partnership, Limited Partnership, Limited Company, Joint Stock Company, Association, Natural Person, Other |
| Gender | Female, Male, NA (for the enterprise or in the absence of data for a natural person) |
| Age | Age of the debtor or their legal representative at the time the debt was incurred, rounded to full years or NA (in the absence of data) |
| Region | Geographical region: 10 regions of Poland and NA (in the absence of data) |
| Phone call | Completed phone call: Yes, No |
| email | Email sent: Yes, No |
| Letter | Postal letter sent: Yes, No |
| SMS | SMS sent: Yes, No |
| Collection stage | Recovery stage: Amicable, Enforcement |

* approximate exchange rate: 4 PLN = 1 USD.

### 4.2. Building the Model

Constructing decision trees requires considering not only the splitting criterion but also the selection of dependent variables for the model. Natural candidates for the splitting criterion are variables dividing the stages of the debt collection process and variables indicated by experts (e.g., gender, place of residence, legal form, and debt class). Such splitting criteria were used in the initial stage of constructing classification trees. As it turned out, some of these attributes proved to be insignificant, partially questioning the opinions expressed by the experts. Equally natural is the use of repayment class as the dependent variable, understood as the ratio of the sum of payments to the nominal value of the debt. With the goal of constructing a classification tree characterized by a high degree of classification accuracy, several possible divisions into repayment classes were experimentally verified.

Another challenge in building decision trees is their susceptibility to overfitting. Hence, the decision tree cannot be too complex, as it will be overfitted and unable to generalize. On the other hand, a too simple tree with a high level of generalization will create too few and too obvious decision rules.

For the construction of classification trees and evaluation of classification accuracy, the machine learning module of the TIBCO Statistica® package version 13.3 x64 [40] was used, employing the mechanism of V-fold cross-validation. In the study, we adopted the parameter V = 10. Initially, we divided the data into four repayment classes, resulting in a total of eight repayment classes when considering the stages of debt collection. Assignments to classes were made based on the relationship between the amount of repayment received and the original debt amount. The defined classes along with the threshold for assignment to each class and the group code are presented in Table 2.

Adopting eight debt repayment classes (case one) led to the construction of a classification tree that almost entirely classified into two extreme classes—full repayment or non-repayment. The accuracy of the decision tree classification for the eight repayment classes is presented in tabular form in Appendix A in Table A6.

Leaving eight repayment classes unchanged, when only extreme classes are recognized and cases assigned to the Good Repayment and Low Repayment classes are completely misclassified, was not acceptable. In the next step, we divided the receivables into six repayment classes, taking into account the stage of debt collection. These classes were: Very Good Repayment, Good Repayment, and Low Repayment (each in one of two debt collection stages: amicable and enforcement). This division is presented in Table 3.

**Table 2.** Repayment Classes—Initial Split.

| Repayment Class | Collection Stage | Total Payments towards the Amount Owed [%] |
|---|---|---|
| Full repayment | Amicable debt collection<br>Enforced debt collection | 100 |
| Good repayment | Amicable debt collection<br>Enforced debt collection | [35–100) |
| Low repayment | Amicable debt collection<br>Enforced debt collection | (0–35) |
| Non-repayment | Amicable debt collection<br>Enforced debt collection | 0 |

**Table 3.** Repayment Classes—Second Attempt at Division.

| Repayment Class | Collection Stage | Total Payments towards the Amount Owed [%] |
|---|---|---|
| Very good repayment | Amicable debt collection<br>Enforced debt collection | ≥75 |
| Good repayment | Amicable debt collection<br>Enforced debt collection | [35–75) |
| Low repayment | Amicable debt collection<br>Enforced debt collection | <35 |

Once again, we obtained classification for two extreme classes with almost complete omission of the Good Repayment class. Both of the above cases led to the observation that correct classification was possible into two distinct repayment classes, a situation also confirmed in the literature [41]. The binary division into the Good Repayment and Low Repayment classes simultaneously constituted a natural division for the classification and regression tree (C&RT) we used. Ultimately, we adopted a division into two repayment classes and two debt collection stages within each of them.

Another particularly important task turned out to be assigning debts to repayment classes based on the level of debt recovery. When proposing the allocation of receivables to a given repayment class, one must primarily consider the cost of acquiring the receivables and the level of direct debt collection costs. We proposed five potential ways of allocating receivables to repayment categories, as presented in Table 4.

**Table 4.** Proposed methods of classifying receivables by repayment level.

| Variant | A | B | C | D | E |
|---|---|---|---|---|---|
| Repayment Class | Splitting Criterion | | | | |
| Good repayment | >25% | >35% | >45% | >75% | Total payments > debt prices |
| Low repayment | [0–25)% | [0–35)% | [0–45)% | [0–75)% | Total payments ≤ debt prices |

Adopting a repayment criterion at the 25% level of the original receivables value assumed covering the average purchase price of the receivables package (calculated based on the source data). Choosing the 35% level as the splitting criterion was based on the fact that, at this level, the sum of repayments covered the average purchase price of the receivables as well as 50% of the direct costs of conducting the recovery process. Setting the 45% level as the criterion was dictated by the fact that, at this level, the sum of repayments to the original receivables value covered the average purchase price as well as the average direct costs of conducting recovery activities, amounting to approximately 20% of the debt. Repayments at the 75% level of the nominal receivables value covered the average purchase

cost, direct service costs, indirect costs, and the expected profit of the debt collection entity. The division according to variant E was introduced as a control against case A (division at the 25% level). Variant E was based on a real comparison of repayments with the purchase amount of the receivables, rather than the assumed average purchase price. For the proposed five variants, classification trees were constructed, and the accuracy of classification was estimated. A summary of the results of experiments with different division variants into repayment classes is presented in Table 5.

**Table 5.** Classification accuracy levels for four repayment classes, all variants.

| | | Variant A | Variant B | Variant C | Variant D | Variant E |
|---|---|---|---|---|---|---|
| **Repayment Class** | **Collection Stage** | Accuracy [%] | | | | |
| Good Repayment | Amicable debt collection | 86.9 | 86.8 | 87.0 | 87.3 | 86.6 |
| | Enforced debt collection | 78.0 | 80.7 | 82.3 | 82.9 | 78.7 |
| Low Repayment | Amicable debt collection | 79.9 | 79.9 | 81.2 | 79.7 | 80.0 |
| | Enforced debt collection | 88.6 | 87.6 | 87.2 | 85.7 | 88.6 |

Ultimately, after a series of experiments, we decided to classify all receivables into the Low Repayment category in cases where the outcomes of the recovery process did not cover the receivables purchase price and the average direct costs of the recovery process. Receivables would be classified as Good Repayment if the total repayments, regardless of the stage of the recovery process, covered at least the purchase price of the receivables package and the average cost of recovery. This approach corresponded to variant C and allowed for covering the costs of purchasing receivables and the direct costs of the recovery process. The new classification into repayment classes included:

- Good Repayment Amicable Recovery (GoodAmicable)—total repayments greater than the receivables purchase price and average direct cost of recovery.
- Good Repayment Enforced Recovery (GoodEnforced)—total repayments greater than the receivables purchase price and average direct cost of recovery.
- Low Repayment Amicable Recovery (LowAmicable)—total repayments less than or equal to the receivables purchase price and average direct cost of recovery.
- Low Repayment Enforced Recovery (LowEnforced)—total repayments less than or equal to the receivables purchase price and average direct cost of recovery.

## 5. Results

For the adopted variant of dividing debt repayment classes, classification trees were rebuilt using the V-fold validation test mechanism and a randomly split dataset into a training set and a validation set in 2/3 and 1/3 ratios, respectively. The overall achieved classification accuracy is summarized in Table 6.

**Table 6.** Comparison of classification accuracy levels.

| | | v-Fold Cross-Validation | Training Set | Validation Set |
|---|---|---|---|---|
| **Repayment Class** | **Collection Stage** | Accuracy [%] | | |
| Good Repayment | Amicable debt collection | 87.0 | 87.0 | 87.0 |
| | Enforced debt collection | 82.3 | 82.2 | 82.5 |
| Low Repayment | Amicable debt collection | 81.2 | 81.3 | 81.0 |
| | Enforced debt collection | 87.2 | 87.2 | 87.1 |

The accuracy of the constructed decision tree for classifying debts into one of the four repayment classes remained at a very high level in each class, ranging from 82.2% to 87.2%. This was an important argument in favor of using a decision tree in the debt classification process. However, to check whether such a high level of classification accuracy was not

due to the specificity of the dataset and whether it could be achieved using other machine learning methods, we conducted a comparison with other methods:

- Support Vector Machine (SVM)—model parameters: random assignment to the training and test sets, maintaining a 75% proportion of the dataset for the training set and the remaining 25% for the test set. Kernel type: linear and RBF (radial basis function). For both types of kernels, 1000 iterations were performed.
- Neural Network (SNN)—random sampling was adopted, with the following sample sizes: 70% for the training set, 15% for the test set, and 15% for the validation set. The considered type of network was MLP (from 4 to 12 hidden layers), with the following activation functions: linear, logistic, hyperbolic tangent, exponential, and sinusoidal. The sum of squares error or mutual entropy was used as the error function.
- k-Nearest Neighbor (k-NN)—parameters: random allocation to the training set with a size of 75% of cases; Euclidean distance measure.

For each of the considered comparison methods, the same dataset was used as for the decision tree. The TIBCO Statistica® package was used again to build the models.

The average classification accuracies obtained using the decision tree and the selected comparative methods are summarized in Table 7.

**Table 7.** Average classification accuracies of selected machine learning methods.

| Classification Method | Accuracy [%] |
|---|---|
| Support Vector Machine (SVM) | 59.0 |
| Artificial Neural Network (ANN) | 53.8 |
| k-Nearest Neighbor (k-NN) | 65.3 |
| Decision Tree | 85.5 |

The decision tree achieved the highest classification accuracy among the machine learning methods tested in the automatic learning process. Based on this, the decision to use a decision tree as an effective tool for generating rules for dealing with purchased mass debt portfolios was made, regardless of the fact that artificial neural networks and support vector machines are unable to generate decision rules in an explicit form.

The construction of the classification tree was based on several attributes describing the characteristics of debts, creditors, and actions that can be taken in the debt collection process. In the process of building the classification tree, its depth was adjusted by changing the parameters determining the minimum number of observations in a leaf and the proportion of observations from a given class in the total number of observations in a leaf from 0.5 to 0.65. By comparing the classification accuracy on the training and test sets, we aimed to achieve comparable classification quality results indicative of good tree generalization with no overfitting. After a series of experiments, it was assumed that the proportion of observations from the dominant class in a leaf must be no less than 65% for the leaf to be classified into a specific class.

After setting the threshold for assigning a leaf to one of the four classes at 65% of observations from that class, we proceeded with selecting the optimal tree based on criteria of tree construction costs and misclassification costs. The final tree was selected from a sequence of 17 classification trees representing subsequent stages of tree building and pruning. The sequential tree construction involved trees containing from 29 leaves to 1 leaf. The criterion for selecting the tree was comparing the cost of cross-validation, which increased with the growth of the tree, to the cost of resubstitution (error rate), which decreased for a smaller tree [42]. The accuracy of the final classification tree was 86.4%. Detailed accuracy measures for the final classification tree are presented in Table 8. We used classic multiclass confusion matrices and metrics [43].

**Table 8.** Classification accuracy of the final decision tree for all classes.

| | Low Enforced | Low Amicable | Good Amicable | Good Enforced | Macro Average |
|---|---|---|---|---|---|
| Sensitivity | 0.92 | 0.76 | 0.92 | 0.75 | 0.83 |
| Precision | 0.87 | 0.82 | 0.89 | 0.84 | 0.86 |
| F1 | 0.90 | 0.79 | 0.90 | 0.79 | 0.85 |

The final decision tree is presented in the attached Figure A1 and Table A7 in Appendix A. The chosen decision tree consisted of 7 levels and 16 end nodes (leaves) labeled with unique identifiers. An assessment of predicate importance was conducted for the final decision tree. The obtained results are presented in Table 9.

**Table 9.** Predicate Importance.

| Predicate | Validity |
|---|---|
| Legal form | 0.56 |
| Phone call | 0.56 |
| Letter | 0.46 |
| Collection stage | 0.46 |
| Region | 0.40 |
| Gender | 0.39 |
| Email | 0.15 |
| Age | 0.06 |

Heading toward developing a set of rules based on the decision tree structure, we formulated rules representing successive splits from the root to each leaf. As a result of these actions, we obtained 16 rules, one for each leaf. The leaf numbers did not correspond to the rule numbers and were the result of continuous numbering during the sequential construction of subsequent trees. The set of obtained rules is presented in Table 10. The structure of observations in each leaf is presented in Table A8 in Appendix A. The compilation included information regarding the number of cases classified in each leaf into one of the considered repayment classes, taking into account the stage of debt collection. The obtained values indicated the proportion of the dominant class in each leaf of the analyzed decision tree.

**Table 10.** Set of all rules extracted from the decision tree.

| Rule ID (Leaf Number) | Premises IF | Conclusion THEN |
|---|---|---|
| 01 (129) | The debt is above PLN 500 and a letter was sent to a woman (or no data available) residing outside the area (Lublin, Kielce, Krakow, Rzeszow, Warsaw, Olsztyn, or Bialystok). | Low repayment at the enforcement stage of debt collection |
| 02 (135) | The debt is between PLN 500 and 2000 and a letter was sent to a woman residing in one of the areas (Lublin, Kielce, Krakow, Warsaw, Olsztyn, or Bialystok), not engaged in business activities or engaged in activities in legal forms such as sole proprietorship, civil partnership, or association. | Good repayment at the enforcement stage of debt collection |
| O3 (134) | The debt is in the range from PLN 500 to 2000 and a letter was sent to a woman engaged in business activities in one of the areas (Lublin, Kielce, Krakow, Rzeszow, Warsaw, Olsztyn, or Bialystok), under the legal form of general partnership, limited liability company, joint-stock company, or limited partnership. | Low repayment at the enforcement stage of debt collection |

**Table 10.** *Cont.*

| Rule ID (Leaf Number) | Premises IF | Conclusion THEN |
|---|---|---|
| 04 (130) | The debt is in the range from PLN 500 to 2000 and a letter was sent to a man residing in one of the areas (Lublin, Kielce, Krakow, Rzeszow, Warsaw, Olsztyn, or Bialystok). | Low repayment at the enforcement stage of debt collection |
| 05 (108) | The debt is above PLN 500 and a letter (payment demand) was sent to a man. | Low repayment at the enforcement stage of debt collection |
| 06 (63) | The debt is up to PLN 500 and a payment demand was sent to a woman. | Good repayment at the enforcement stage of debt collection |
| 07 (81) | The debt is up to PLN 500, telephone contact was made, and a payment demand was sent to a man residing outside the area (Lublin, Kielce, or Warsaw), conducting business activities in one of the legal forms (sole proprietorship, joint-stock company, civil partnership, or general partnership). | Good repayment at the enforcement stage of debt collection |
| 08 (80) | The debt is up to PLN 500, telephone negotiations were conducted, and a payment demand was sent to a man not engaged in business activities or engaged in business activities in one of the legal forms (limited liability company, limited partnership, or association), residing/conducting business activities outside the area (Lublin, Kielce, or Warsaw). | Low repayment at the enforcement stage of debt collection |
| 09 (76) | The debt is up to PLN 500, telephone negotiations were conducted, and a payment demand was sent to a man residing outside the area (Lublin, Kielce, or Warsaw). | Good repayment at the enforcement stage of debt collection |
| 10 (67) | The debt is up to PLN 500 and a payment demand was sent to entities conducting business in one of the areas (Lublin, Kielce, or Warsaw). | Low repayment at the enforcement stage of debt collection |
| 11 (66) | The debt is up to PLN 500 and a letter was sent to a man residing in the area (Lublin, Kielce, or Warsaw). | Good repayment at the enforcement stage of debt collection |
| 12 (5) | The debt is above PLN 2000 and no payment demand was sent. | Low repayment at the amicable debt collection stage |
| 13 (21) | The debt is up to PLN 500 or above PLN 2000 and a payment demand was sent via email. | Low repayment at the amicable debt collection stage |
| 14 (20) | The debt is up to PLN 500 and no payment demand was sent via email. | Good repayment at the amicable debt collection stage |
| 15 (9) | The debt is from PLN 500 to 2000 and telephone negotiations were conducted. | Good repayment at the amicable debt collection stage |
| 16 (8) | The debt is from PLN 500 to 2000 and no telephone contact was made. | Low repayment at the amicable debt collection stage |

Approximate exchange rate: 4 PLN = 1 USD.

## 6. Discussion

The obtained set of 16 rules indicated for which amount of debt and under what debtor characteristics selected actions should be taken within the scope of amicable or compulsory debt collection. The rules also specified for which types of debt no actions should be taken due to minimal chance of recovery. For example, rule 1 indicated that debts above PLN 500 from women residing in specified regions of Poland should not be directed to legal proceedings, as they do not promise repayment (and additionally expose the debt collection company to legal costs). On the other hand, rule 15 indicated that if the debt is above PLN 500 but below PLN 2000 and contact can be established via phone, good repayment during the amicable collection stage is possible.

The set of rules developed using machine learning methods constitutes a ready-made solution introducing a division of debts enabling their direction to either the amicable or compulsory collection stage. At the same time, by indicating the effect of the actions taken, the rules create an opportunity to develop debt management strategies. The obtained rules can be directly implemented in the information systems of the debt collection entity using generated SQL code or a set of decision rules. An example of such a system is the REBIT rule-based system [44] utilizing a relational database as a tool for collecting business rules. The presented concept of a debt management model based on a set of procedural rules

highlights the very significant impact of debt classification based on their characteristic features on the way they are handled in the debt collection process.

Determining the number of debt classes and the required debt recovery level proved to be a crucial element in data preparation, upon which the subsequent accuracy of the model and its ability to generate stable procedural rules depended. This required numerous experiments, the results of which may differ for other types of debts than the mass debts we analyzed. However, it turned out that, with a greater number of classes, the decision tree algorithm mainly classified into two extreme classes.

The obtained set of rules was presented to six experts, who positively evaluated it and proposed an additional 16 expert rules. They mainly concerned the organization of debt collection teams and their activities. We do not present these rules, as they do not directly result from the available dataset. However, the expert rules indicated that, in order to obtain an effective automatic tool for building a set of procedural rules in the debt collection process, it is necessary to supplement the dataset with additional attributes. These attributes should include detailed information transferred from debt collectors' notes regarding contacts with the debtor, as well as data considering the frequency of contact with the debtor through various methods, such as in [17].

Our research has introduced several important contributions to the literature. Firstly, we have collected a large dataset of real-world data and demonstrated the possibility of extracting knowledge from data collected in the debt recovery process leading to the creation of debt recovery procedures. To our knowledge, there has been no such research to date.

Secondly, we have built a set of rules for debt collection procedures based on machine learning methods. Previous research has focused mainly on determining the level of recovered debt amounts [2,12,16]. Only research in [17] focused on the debt collection process itself, but it only considered the problem of optimizing the frequency of phone calls made to debtors. Similarly, research in [15] focused on phone calls and obtaining a promise to repay the debt. This research problem belongs to the general area of multi-criteria optimization [45].

Thirdly, we have defined a methodology for extracting rules for debt collection procedures, showing the need to divide debts into only two repayment classes, thus confirming the results of a previous study [46]. We also demonstrated the effectiveness of decision trees as a tool for generating decision rules for the debt collection process.

Fourthly, we have discovered the possibility of reducing certain activities in the debt collection process, depending on the characteristics of the debt and the debtor. This will increase the efficiency of the debt collection process and limit ineffective contact with the debtor, which will reduce job burnout among employees of debt collection companies.

Based on the conducted research, we proposed a general model of the debt collection process management for bulk debts. The schematic model is drawn at a very high level of generality and is one of many that can be built based on the developed rule base. It is illustrated by a simplified BPMN diagram in Figure 3. Particularly important in our proposed model of the debt collection process is that, as a result of data collection during subsequent debt collection processes, the rule base can be updated. The continuously updated rule database can be used to automate the debt valuation and acquisition process, and then to support decisions about the selection of procedures for handling the acquired debts. The use of data from the collection of new debt portfolios in the model not only enriches the rule base but also makes the model adaptive, adjusting to the specifics of a given debt collection entity, as illustrated by the arrows leading to the rule base in Figure 3.

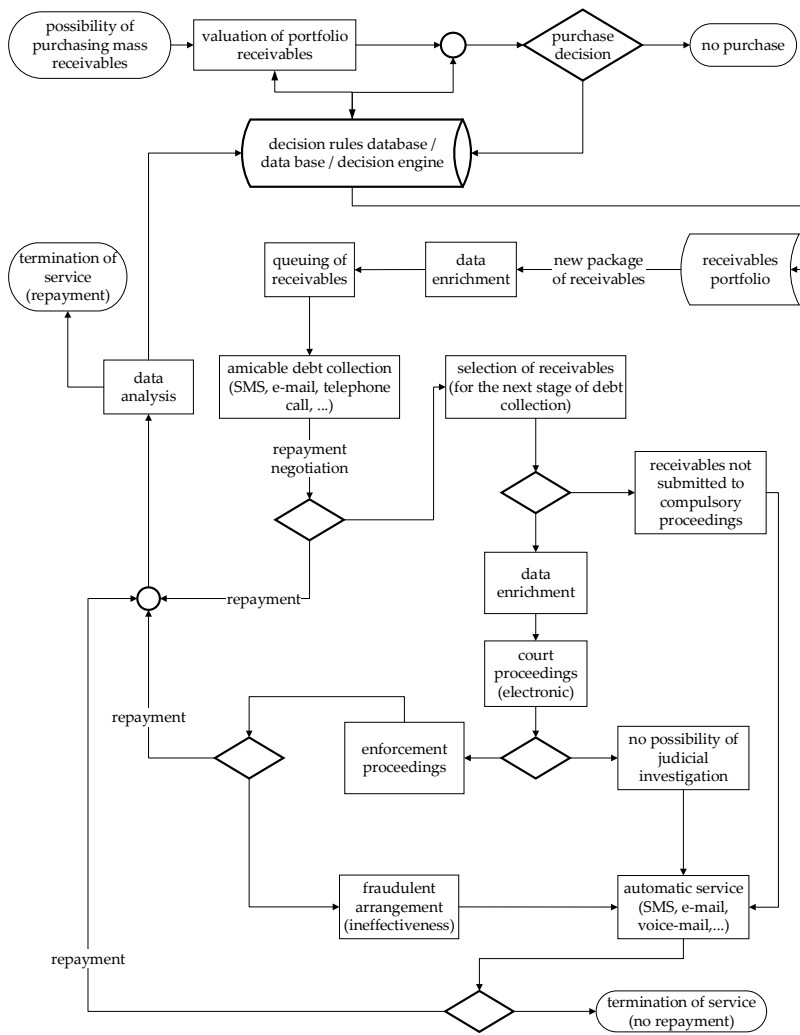

**Figure 3.** Model of managing the mass debt collection process.

## 7. Conclusions

Our research confirmed the possibility of generating rules for managing mass debt collection processes based on a dataset from a debt collection company. In the case of mass debts, a large dataset with a similar data structure allows for the utilization of a decision tree algorithm to create a set of business rules. Attempting to use associated methods did not yield satisfactory results.

The dataset, comprising over 800,000 records on debts, related activities, and outcomes of the debt collection process, underwent preprocessing and discretization, enabling the creation of 16 procedural rules using a decision tree. A significant challenge in preparing the data for inference was deciding on the number of target repayment classes and the debt recovery level assigned to each class. Ultimately, we divided the data into two classes of low and high repayment, where high repayment indicated covering the purchase price of the debt. The classification accuracy of the final decision tree version was 85.5%. The attributes that had the strongest impact on the debt collection process outcome were the debtor's legal form, initiating a phone call to the debtor, and sending a payment reminder letter.

We successfully addressed the research questions Q1–Q3 posed at the beginning. We managed to create a set of non-trivial decision rules for the debt collection process, providing new insights into the selection of the appropriate process depending on the characteristics of the debt and debtor. Question Q4 remains unresolved at this stage due to the limited dataset concerning debtor characteristics and detailed descriptions of the debt collection process.

A decision rule-based model will improve the efficiency of collection agency employees' actions and reduce excessive contact with debtors in unpromising situations. The sense of greater effectiveness and the reduction of psychologically taxing interactions with difficult debtors will help mitigate burnout and ensure sustainable employee development, thereby extending the currently short employment duration in collection agencies.

The main limitation of the study was the relatively limited set of predictors that could be obtained based on available data from debt collection companies. Further work should primarily involve expanding the information systems of debt collection companies to collect detailed information about actions taken regarding each debt under collection.

Future research should focus on expanding the set of attributes, allowing for the creation of a more extensive and detailed set of rules. Such a rule set could then be used for automatic debt purchase and to support the management of bulk debts using an inference engine (e.g., Business Rules Management System "REBIT" [44,47]). Only then will it be possible to answer research question Q4 regarding the possibility of automating the debt purchase and collection process.

Our research has many potential practical applications, such as automating the debt collection process for low nominal value debts, reducing the number of phone calls made, and decreasing the number of debts taken to court by implementing mechanisms for selecting debts that are unlikely to be repaid, even at the forced collection stage.

During the research, we noticed the potential to use datasets to build a recommendation system for individual debt collection strategies tailored to debtor characteristics. Such actions will require further research to determine the relationships and effectiveness of applied debt collection techniques (negotiation strategies) concerning specific debtor groups.

**Author Contributions:** Conceptualization, R.J. and A.P.; methodology R.J. and A.P.; software, R.J. and A.P.; validation, R.J. and A.P.; formal analysis, R.J. and A.P.; investigation, R.J. and A.P.; resources, R.J. and A.P.; data curation, R.J. and A.P.; writing—original draft preparation, R.J. and A.P.; writing—review and editing, R.J. and A.P.; visualization, R.J. and A.P.; supervision, R.J. and A.P.; project administration, R.J. and A.P.; funding acquisition, R.J. and A.P. All authors have read and agreed to the published version of the manuscript.

**Funding:** The APC was funded under subvention funds for the Faculty of Management of the AGH University of Krakow.

**Institutional Review Board Statement:** Not applicable.

**Informed Consent Statement:** Not applicable.

**Data Availability Statement:** The datasets presented in this article are not readily available due to legal limitations.

**Conflicts of Interest:** The authors declare no conflicts of interest.

## Appendix A

**Table A1.** Division of receivables into classes according to the nominal debt value in PLN [1].

| Debt Class | Quantity | Percentage Share [%] |
|---|---|---|
| (0, 500] | 384,158 | 43.7 |
| (500, 1k] | 105,087 | 12.0 |
| (1k, 2k) | 126,209 | 14.4 |
| (2k, 5k] | 154,221 | 17.5 |
| (5k, 10k] | 63,064 | 7.2 |
| (10k, 25k] | 30,756 | 3.5 |
| (25k, 50k] | 12,632 | 1.4 |
| (50k,100k] | 2186 | 0.2 |
| (100k, 250k] | 636 | 0.1 |
| (250k, 500k] | 40 | 0.0 |
| (500k, 1000k] | 18 | 0.0 |
| Total | 879,007 | 100.0 |

[1] approximate exchange rate: 4 PLN = 1 USD.

**Table A2.** Division of receivables according to the legal form of the debtor.

| Legal Form | Quantity | Share [%] |
|---|---|---|
| Business activity (sole proprietorship) | 263,094 | 29.9 |
| Joint-Stock company (JSC) | 1737 | 0.2 |
| Partnership | 6668 | 0.8 |
| General partnership (GP) | 1759 | 0.2 |
| Limited partnership (LP) | 572 | 0.1 |
| Limited Liability Company (LLC) | 23,557 | 2.7 |
| Association | 304 | 0.0 |
| Individual (natural person) | 581,036 | 66.1 |
| Other | 280 | 0.0 |
| Total | 879,007 | 100.0 |

**Table A3.** Number of receivables by method of communication with the debtor.

| Attribute | Outbound Call | Sent Email | Sent Letter |
|---|---|---|---|
| YES (True) | 732,111 | 39,461 | 540,993 |
| NO (FALSE) | 146,896 | 839,546 | 338,014 |

**Table A4.** Distribution of receivables by debt collection method.

| Name | Quantity | Share [%] |
|---|---|---|
| Amicable debt collection | 338,014 | 38.5 |
| Enforced debt collection | 540,993 | 61.5 |

**Table A5.** Basic statistics of selected variables before discretization.

| Name | Average | Standard Deviation | Minimum | Maximum |
|---|---|---|---|---|
| Principal debt [PLN] | 2647 | 8162 | 0.3 | 995,514 |
| Age [years] | 45 | 14 | 19 | 98 |
| Number of phone calls | 38 | 69 | 0 | 285 |
| Number of letters sent | 1 | 0.7 | 0 | 6 |
| Number of emails sent | 1 | 0.2 | 0 | 6 |

**Table A6.** Classification accuracy levels for eight repayment classes.

| | | | Predicted Repayment Class [%] | | | | | | | |
|---|---|---|---|---|---|---|---|---|---|---|
| | | | Full Repayment | | Good Repayment | | Low Repayment | | Non-Repayment | |
| | | | Amicable debt collection | Enforced debt collection | Amicable debt collection | Enforced debt collection | Amicable debt collection | Enforced debt collection | Amicable debt collection | Enforced debt collection |
| Observed repayment class | Full repayment | Amicable debt collection | 87.3 | | | | | | | |
| | | Enforced debt collection | | 84.0 | | | | | | |
| | Good repayment | Amicable debt collection | | | 0.0 | | | | | |
| | | Enforced debt collection | | | | 0.0 | | | | |
| | Low repayment | Amicable debt collection | | | | | 0.0 | | | |
| | | Enforced debt collection | | | | | | 8.9 | | |
| | Non-repayment | Amicable debt collection | | | | | | | 79.6 | |
| | | Enforced debt collection | | | | | | | | 83.4 |

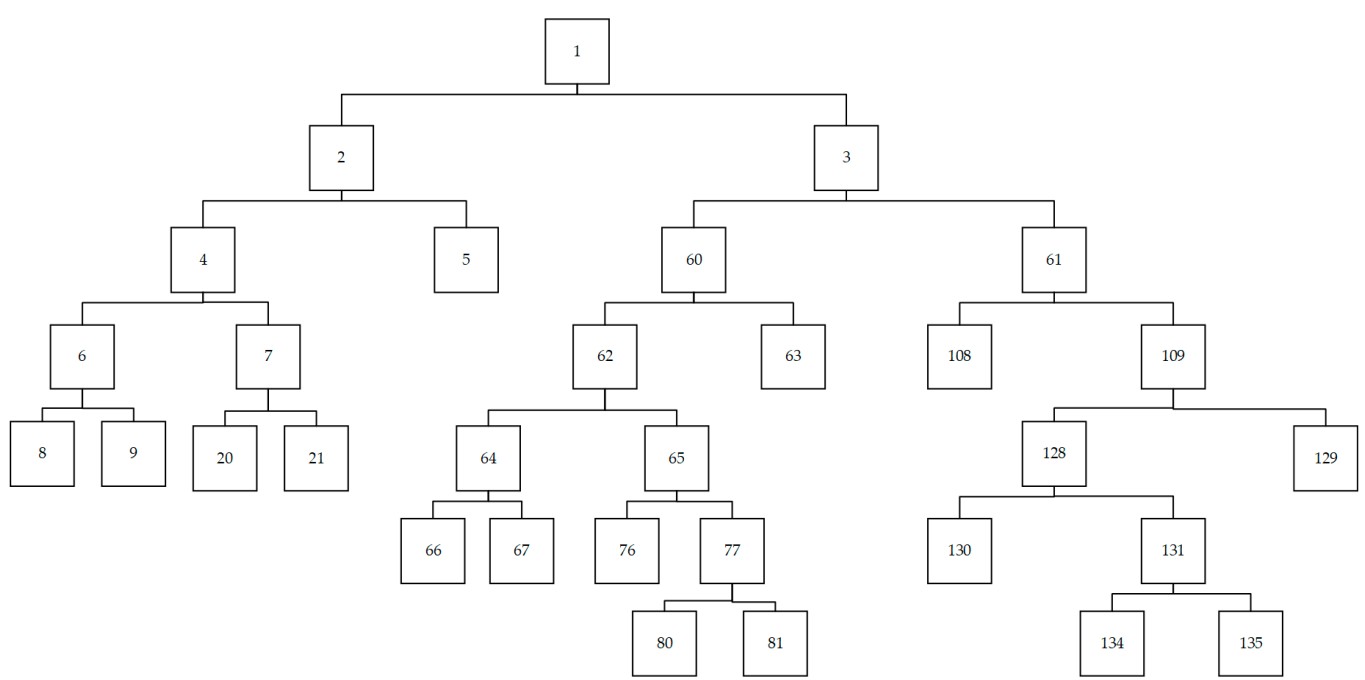

**Figure A1.** Final decision tree (15 nodes, 16 leaves).

**Table A7.** Conditions for splitting nodes of final decision tree.

| Node | Left | Right | Dominant Class | Division (Variable) | Value |
|------|------|-------|----------------|---------------------|-------|
| 1 | 2 | 3 | LowEnforced | Letter | NO |
| 2 | 4 | 5 | GoodAmicable | Account class | (0, 500], (500, 1k], (1k, 2k) |
| 4 | 6 | 7 | GoodAmicable | Account class | (500, 1k], (1k, 2k) |
| 6 | 8 | 9 | GoodAmicable | Phone call | NO |
| 8 | | | LowAmicable | | |
| 9 | | | GoodAmicable | | |
| 7 | 20 | 21 | GoodAmicable | email | NO |
| 20 | | | GoodAmicable | | |
| 21 | | | LowAmicable | | |
| 5 | | | LowAmicable | | |
| 3 | 60 | 61 | LowEnforced | Account class | (0, 500] |
| 60 | 62 | 63 | GoodEnforced | Gender | Male, NA |
| 62 | 64 | 65 | LowEnforced | Region | Lubelskie, Kieleckie, Warszawskie |
| 64 | 66 | 67 | GoodEnforced | Gender | Male |
| 66 | | | GoodEnforced | | |
| 67 | | | LowEnforced | | |
| 65 | 76 | 77 | LowEnforced | Phone call | NO |
| 76 | | | GoodEnforced | | |
| 77 | 80 | 81 | LowEnforced | Legal form | Natural person, LLC, LP, Assoc., Other |
| 80 | | | LowEnforced | | |
| 81 | | | GoodEnforced | | |
| 63 | | | GoodEnforced | | |
| 61 | 108 | 109 | LowEnforced | Gender | Male |
| 108 | | | LowEnforced | | |
| 109 | 128 | 129 | LowEnforced | Region | Lubelskie, Kieleckie, Krakowskie, Rzeszowskie, Warszawskie, Olsztynskie, Bialostockie |

**Table A7.** *Cont.*

| Node | Left | Right | Dominant Class | Division (Variable) | Value |
|------|------|-------|----------------|---------------------|-------|
| 128 | 130 | 131 | LowEnforced | Account class | (2k, 5k], (5k, 10k], (10k, 25k], (25k, 50k], (50k,100k], (100k, 250k], (250k, 500k] |
| 130 | | | LowEnforced | | |
| 131 | 134 | 135 | GoodEnforced | Legal form | GP, LLC, LP, JSC, Assoc., Other |
| 134 | | | LowEnforced | | |
| 135 | | | GoodEnforced | | |
| 129 | | | LowEnforced | | |

LowAmicable—Low repayment at the amicable debt collection stage. GoodAmicable—Good repayment at the amicable debt collection stage. LowEnforced—Low repayment at the enforcement stage of debt collection. GoodEnforced—Good repayment at the enforcement stage of debt collection.

**Table A8.** Count of observations in the terminal node (leaf).

| Leaf | Class LowAmicable | Class GoodAmicable | Class LowEnforced | Class GoodEnforced |
|------|-------------------|--------------------|-------------------|--------------------|
| 8 | 12,594 | 4760 | 0 | 0 |
| 9 | 7854 | 28,356 | 0 | 0 |
| 20 | 13,991 | 164,739 | 0 | 0 |
| 21 | 3312 | 1544 | 0 | 0 |
| 5 | 78,368 | 22,496 | 0 | 0 |
| 66 | 0 | 0 | 6512 | 22,016 |
| 67 | 0 | 0 | 792 | 481 |
| 76 | 0 | 0 | 4296 | 12,155 |
| 80 | 0 | 0 | 39,987 | 5056 |
| 81 | 0 | 0 | 4376 | 8694 |
| 63 | 0 | 0 | 22,183 | 74,024 |
| 108 | 0 | 0 | 181,424 | 13,262 |
| 130 | 0 | 0 | 20,431 | 2068 |
| 134 | 0 | 0 | 758 | 171 |
| 135 | 0 | 0 | 10,493 | 21,881 |
| 129 | 0 | 0 | 81,120 | 8813 |

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
