# Peer review of "Debt Collection Model for Mass Receivables Based on Decision Rules—A Path to Efficiency and Sustainability"

_sustainability, doi:10.3390/su16145885_

Round 1

Reviewer 1 Report

Comments and Suggestions for Authors

Comments on the Quality of English Language

Author Response

Thank you for all of your comments and suggestions. We have tried to incorporate most of them.

  1. What is the merit of the proposed machine learning methods?

In the course of our research, we analyzed the possibility of applying various machine learning algorithms, including artificial neural networks, SVM, k-NN, and decision trees. We chose decision trees for generating rules due to their high accuracy in each of the considered classes, as well as their advantages such as: ease of generating transparent decision rules, the ability to present the entire decision-making process in the form of a tree, and a relatively short computation time for such a large dataset as we were able to collect.

  1. The main motivation of this paper shall be improved in Introduction.

We have introduced in the Introduction additional reasons for conducting research in the article.

  1. Please give more explanations on the theoretical model of the debt collection

process, see Figure 1.

We have added an additional paragraph of text to the description of Figure 1.

  1. Compared with the previous works, what is the advantage of the theoretical

model of the debt collection process?

We have added an additional paragraph of text to the description under Figure 1.

  1. Please give more explanations on Figure 3 in the text.

In the explanation of Figure 3, we have added the point that using feedback to update the rule base leads to the creation of a model that adapts to the specifics of a given collection entity.

  1. Please state the practical application of the obtained results in this paper in Conclusion section.

The Conclusion section has been expanded to include, among other things, an indication of practical applications.

  1. The following works play a vital role in machine learning and artificial intelligence. Please cite them.

We have cited both works.

Reviewer 2 Report

Comments and Suggestions for Authors

The article submitted for review is devoted to the issue of increasing the efficiency of repayment of borrowed funds carried out by a special company from individuals.

The article contains all the main required sections - literature review, methods, results analysis, discussion of results and conclusions. There are also several appendices to the main text.

The references corresponds to the topic.

However, there are some comments over the work aimed solely at improving the quality of presentation of the results :

1. The title is too cumbersome and, in my opinion, it needs to be shortened, focusing on machine learning methods.

2. The article deals with accounts receivable of individuals, Fig. 2 shows the factors characterizing both individuals and companies. It would be better to list only the factors of those receivables to which this article is devoted - that is, individuals.

3. The rule base (table 10) is given after the results of the classification of subjects. I recommend placing it after the discussion of the principles of constructing the rule base, in section 4.2.

4. When comparing the effectiveness of machine learning methods, the authors use only the metric - accuracy. But this metric alone cannot give an idea of ​​the quality of classification. It is necessary to supplement the analysis results with other metrics.

Author Response

Thank you for all of your comments and suggestions. We have tried to incorporate most of them.

  1. The title is too cumbersome and, in my opinion, it needs to be shortened, focusing on machine learning methods.

We shortened the title of the article while striving to preserve its main meaning.

  1. The article deals with accounts receivable of individuals, Fig. 2 shows the factors characterizing both individuals and companies. It would be better to list only the factors of those receivables to which this article is devoted - that is, individuals.

We had to keep the characteristics of enterprises in Figure 2, as individuals running businesses account for nearly 30% of our data, and enterprises account for another 3%.

  1. The rule base (table 10) is given after the results of the classification of subjects. I recommend placing it after the discussion of the principles of constructing the rule base, in section 4.2.

We considered moving the set of rules to section 4.2, but the information on the process of building and the final form of the decision tree that was used to create the rule base is not contained until section 5. Placing the rules earlier would create confusion about how they were obtained.

  1. When comparing the effectiveness of machine learning methods, the authors use only the metric - accuracy. But this metric alone cannot give an idea of the quality of classification. It is necessary to supplement the analysis results with other metrics.

We have added Table 9 containing additional classification quality indicators for multi-class classification, along with additional descriptions and references to the literature.

Reviewer 3 Report

Comments and Suggestions for Authors

In their article, the authors presented a debt collection model for mass receipts in a debt collection company based on decision rules. The purpose of constructing such a model is to make the debt collection process more efficient. The article is written correctly due to editorial requirements. It has the correct structure. Includes purpose, research questions. The study was conducted to answer the questions asked. In the Conclusions, the authors addressed these questions. Main notes:

1. In the Discussion, authors should refer the results of their research to the research of other scientists.

2. In what units is the Account class variable given (lines 448-450)? PLN or USD? This is not clear at the moment. Please add this.

3. I have doubts about providing monetary amounts in PLN. Or maybe it would be better to give them in USD or Euro? The adopted exchange rate appeared in the text: PLN 4 = USD 1. However, it is only in one place and readers from outside Poland may have problems reading the information contained in the text. If the authors decide to leave PLN, I suggest providing this conversion rate under the tables and at the end of text fragments where monetary values are given.

4. The research period covers the years 2006-2022. Time has a very large impact on the value of money. The value of the zloty has changed over the years. Have the given values been discounted, e.g. by the inflation rate? Are these the nominal values of the debt? In my opinion, they should be discounted and related, for example, to the value of PLN in the first analyzed year, 2006. In 2022, the prices of goods and services increased by 60.7% compared to 2006 (https://stat.gov.pl/en/topics /prices-trade/price-indices/price-indices-of-consumer-goods-and-services/yearly-price-indices-of-consumer-goods-and-services-from-1950/).

Author Response

Thank you for all of your comments and suggestions. We have tried to incorporate most of them.

  1. In the Discussion, authors should refer the results of their research to the research of other scientists.

In the final part of Section 6 Discussion, we added text in which we compared our research results with other studies and indicated the new contributions made to the literature (lines 707-725).

  1. In what units is the Account class variable given (lines 448-450)? PLN or USD? This is not clear at the moment. Please add this.
  2. I have doubts about providing monetary amounts in PLN. Or maybe it would be better to give them in USD or Euro? The adopted exchange rate appeared in the text: PLN 4 = USD 1. However, it is only in one place and readers from outside Poland may have problems reading the information contained in the text. If the authors decide to leave PLN, I suggest providing this conversion rate under the tables and at the end of text fragments where monetary values are given.

We left the debt amounts in the original Polish złoty (PLN) currency, as stating them in a foreign currency could imply that the debt is denominated in EUR or USD. However, we added information about the PLN currency and the PLN/USD exchange rate in many places.

  1. The research period covers the years 2006-2022. Time has a very large impact on the value of money. The value of the zloty has changed over the years. Have the given values been discounted, e.g. by the inflation rate? Are these the nominal values of the debt? In my opinion, they should be discounted and related, for example, to the value of PLN in the first analyzed year, 2006. In 2022, the prices of goods and services increased by 60.7% compared to 2006 (https://stat.gov.pl/en/topics /prices-trade/price-indices/price-indices-of-consumer-goods-and-services/yearly-price-indices-of-consumer-goods-and-services-from-1950/).

We considered discounting debt amounts before conducting their research, but we decided against it due to the complexity of the procedure for mass receivables and the lack of a significant impact on the results. We assumed in our research that the debt repayment amount would be compared to the purchase price of the receivables. We found that in 45% of cases, the final installment of the debt was repaid within one year, and in another 22% of cases, within the following year of purchasing the receivable. Thus, most of the total repayment amount for the purchased receivables occurs in the first year after purchase, as earlier installments of debt repaid in subsequent years are also repaid in the first year. Therefore, the discounting process would not significantly change the proportion of recovered amounts to the purchase prices of receivables. This information was added to the article by adding an additional paragraph in section 4.1 (lines 453-459).

Round 2

Reviewer 1 Report

Comments and Suggestions for Authors

This paper can be accepted. 

Reviewer 2 Report

Comments and Suggestions for Authors

The article has been corrected and supplemented in accordance with the reviewer's comments.

The manuscript has been sufficiently improved to warrant publication in Sustainability.

Reviewer 3 Report

Comments and Suggestions for Authors

The article has been corrected in accordance with the comments included in the review.